# DNA methylation at the suppressor of cytokine signaling 3 (*SOCS3*) gene influences height in childhood

Prachand Issarapu [1,2], Manisha Arumalla[1], Hannah R. Elliott [3,4], Suraj S. Nongmaithem [1], Alagu Sankareswaran [1,5], Modupeh Betts [2], Sara Sajjadi[1,5], Noah J. Kessler [6], Swati Bayyana[1,5], Sohail R. Mansuri[1,5], Maria Derakhshan[2], G. V. Krishnaveni[7], Smeeta Shrestha [1], Kalyanaraman Kumaran[7,8], Chiara Di Gravio[9], Sirazul A. Sahariah[10], Eleanor Sanderson [3,4], Caroline L. Relton [3,4], Kate A. Ward[2,11], Sophie E. Moore[2,11], Andrew M. Prentice [2], Karen A. Lillycrop[12,13], Caroline H. D. Fall[8], Matt J. Silver [2] ✉, Giriraj R. Chandak [1,5] ✉ & the EMPHASIS study group*

Human height is strongly influenced by genetics but the contribution of modifiable epigenetic factors is under-explored, particularly in low and middle-income countries (LMIC). We investigate links between blood DNA methylation and child height in four LMIC cohorts (*n* = 1927) and identify a robust association at three CpGs in the suppressor of cytokine signaling 3 (*SOCS3*) gene which replicates in a high-income country cohort (*n* = 879). *SOCS3* methylation (*SOCS3m*)−height associations are independent of genetic effects. Mendelian randomization analysis confirms a causal effect of *SOCS3m* on height. In longitudinal analysis, *SOCS3m* explains a maximum 9.5% of height variance in mid-childhood while the variance explained by height polygenic risk score increases from birth to 21 years. Children's *SOCS3m* is associated with prenatal maternal folate and socio-economic status. In-vitro characterization confirms a regulatory effect of *SOCS3m* on gene expression. Our findings suggest epigenetic modifications may play an important role in driving child height in LMIC.

Human height is a complex trait modulated by fixed genetic and variable environmental factors[1]. It is strongly influenced by growth in childhood and short stature is associated with an increased risk of various diseases in adults[2,3]. Multiple studies have investigated the

genetic basis of height and a large number of height-associated variants have been reported[4,5]. In a recent genome-wide association study (GWAS) meta-analysis of 5.4 million adults[6], we identified 12,111 SNPs across different ancestries that together accounted for 90–100% of

[1]Genomic Research on Complex Diseases (GRC-Group), CSIR-Centre for Cellular and Molecular Biology, Hyderabad, Telangana, India. [2]MRC Unit The Gambia at The London School of Hygiene and Tropical Medicine (LSHTM), London, UK. [3]MRC Integrative Epidemiology Unit at the University of Bristol, Bristol, UK. [4]Population Health Sciences, Bristol Medical School, University of Bristol, Bristol, UK. [5]Academy of Scientific and Innovative Research, AcSIR, Ghaziabad, India. [6]Department of Genetics, University of Cambridge, Cambridge, UK. [7]Epidemiology Research Unit, CSI Holdsworth Memorial Hospital, Mysore, Karnataka, India. [8]MRC Lifecourse Epidemiology Centre, University of Southampton, Southampton, UK. [9]Department of Biostatistics, Vanderbilt University Medical Center, Nashville, TN, USA. [10]Centre for the Study of Social Change, Mumbai, Maharashtra, India. [11]Department of Women & Children's Health, King's College London, London, UK. [12]School of Medicine, University of Southampton, Southampton, UK. [13]Biological Sciences, University of Southampton, Southampton, UK. *A list of authors and their affiliations appears at the end of the paper. ✉e-mail: matt.silver@lshtm.ac.uk; chandakgrc@ccmb.res.in

genetic heritability related to height. However, these SNPs explained 40% of adult phenotypic height variance in Europeans and only 10–20% in non-Europeans, suggesting that non-genetic factors may contribute to substantially more height variation in non-Europeans[6]. Evidence from longitudinal studies suggests that the relative genetic contribution increases with age, while environmental exposures show greater impact during early childhood and have long-term consequences[7]. However, the biological mechanisms mediating environmental effects on height during early stages of growth are poorly understood.

Numerous environmental factors are known to influence an individual's growth during childhood, including nutrition[8], infection load[9], socioeconomic status[10], and prenatal maternal nutritional status[11]. For example, children exposed to poor nutritional conditions and high infectious environments are at a higher risk of stunting and manifest associated developmental delays[8,12]. The World Health Organization (WHO) 2021 estimates[13] indicate that a large proportion of stunted children reside in low- and middle-income countries (LMIC), particularly in South Asia and sub-Saharan Africa where undernutrition and associated comorbidities are more prevalent compared to high-income countries (HIC)[14]. This offers a potential explanation for the disparity in height variation attributed to non-genetic factors between LMIC and high-income countries.

Environmental factors can influence a phenotype through their impact on epigenetic processes such as DNA methylation (DNAm) and histone modifications, both of which can influence gene expression[15]. DNAm has been linked to environmental exposures, including nutrition, environmental pollutants, and various prenatal risk factors[16]. Earlier studies on the epigenetic basis of adult height heritability[17] suggest modulation of DNAm patterns at height-associated genes as a candidate mechanism for mediating environmental effects on height. However, there are no genome-wide epigenetic investigations on height in childhood. Such studies may reveal biological mechanisms underlying height and stunting, especially in children from LMIC where environmental effects during growth are greater.

In this study, we perform a discovery epigenome-wide association analysis (EWAS) to identify CpGs where DNAm is associated with height in children from an LMIC cohort (India) and test for replication in three independent LMIC cohorts from India and The Gambia and one HIC cohort from the United Kingdom (UK). Our aim is to (i) identify CpGs associated with child height; (ii) explore genetic and epigenetic contributions to child height variation over time; and (iii) probe the influence of early-life exposures on the establishment of DNAm at height-associated loci. We additionally probe causal pathways within a Mendelian randomization framework and perform in vitro experiments in cell lines to understand the effect of methylation on gene expression at height-associated CpGs. Overall, our study provides strong evidence of genome-wide DNA methylation associations with height in children from LMIC.

## Results
### Study participants, baseline characteristics, and study design
We investigated associations between DNAm and child height in LMIC populations using data from four independent cohorts, two Indian and two Gambian. Given genetic and environmental differences between South Asian and sub-Saharan African populations, we followed a "discovery-replication" study design and started with a discovery EWAS in participants from the Mumbai Maternal Nutrition Project[18] ("MMNP", India) which had the largest sample size ($n = 698$) amongst the LMIC cohorts. Children in this cohort were aged between 5 and 7 years (mean 5.8 years) with a mean height of 109.6 cm [mean height adjusted Z-score (HAZ) −1.01 using the WHO reference[19]]. Replication of loci identified in the discovery EWAS was performed in three separate LMIC cohorts comprising individuals from the Mysore Parthenon Cohort[20] ("MPC", mean age: 5 years, mean HAZ: −0.86, India);

the Periconceptional Multiple Micronutrients Supplementation Trial[18] ("PMMST", mean age: 9 years, mean HAZ: −0.33, The Gambia); and the Early Nutrition and Immune Development trial[21] ("ENID", mean age: 6 years, mean HAZ: −0.66, The Gambia). An additional replication was performed in one HIC cohort from the UK: the Avon Longitudinal Study of Parents and Children[22] ("ALSPAC", mean age: 7.4 years, mean HAZ: 0.37, UK). Further details including baseline characteristics of the cohorts are summarized in Table 1 and an overview of the analysis workflow is provided in Fig. 1.

### Discovery EWAS: *SOCS3* methylation (*SOCS3m*) is associated with child height
A cross-sectional EWAS of 803,210 CpG methylation beta values passing QC on the Illumina EPIC array in the discovery cohort (MMNP) identified significant associations with height at three CpGs: cg11047325 ($P = 3.0 \times 10^{-11}$), cg13343932 ($P = 5.8 \times 10^{-11}$), and cg18181703 ($P = 3.0 \times 10^{-10}$) at a pre-defined false discovery rate threshold, FDR < 0.05 (Table 2, and Fig. 2). These three CpGs mapped to exon 2 of Suppressor of cytokine signaling three gene (*SOCS3*) on chromosome 17. Methylation levels at the three CpGs were strongly correlated (Pearson's $r \geq 0.75$; Supplementary Fig. 1), and the effect sizes were similar, with a 1% increase in methylation associated with an average 0.25 cm increase in height, equivalent to 0.053 SD (Table 2). There was no evidence of genomic inflation of $P$ values ($\lambda = 0.98$; Fig. 2b). Sensitivity analysis revealed no significant changes in effect sizes or $P$ values at associated *SOCS3* CpGs following adjustment for estimated blood cell counts (see Supplementary Data 1) or maternal supplementation (see Supplementary Data 2) indicating that neither influence the observed *SOCS3* methylation (*SOCS3m*)− height association. The same 441 bp region spanning these three CpGs was identified in a differentially methylated region analysis (adjusted $P$ value (Stouffer) = $2.1 \times 10^{-11}$; Supplementary Data 3).

### Replication of *SOCS3m*−height association
We observed a robust replication of the *SOCS3m*−height association in cross-sectional analyses for all three LMIC cohorts in mid-childhood aged between 5 and 9 years, with consistent direction of effects and $P$ values ranging from 0.047 to $3 \times 10^{-14}$ across these different sized cohorts (Table 3). On average, a 1% change in methylation was associated with 0.23, 0.15, and 0.08 cm increase in height, equivalent to 0.054, 0.029, 0.021 SD in the MPC, PMMST, and ENID cohorts, respectively. Similar to the discovery cohort, methylation at these CpGs was correlated in all the replication cohorts (Supplementary Fig. 1). Sensitivity analyses did not identify any effect of cell count variation on the *SOCS3m*−height association in any of the replication cohorts (Supplementary Data 4). We carried out an additional association analysis of early-childhood effects in Gambian ENID participants for which height measurements and methylation data were available at 2 years (mean age: 2 years, mean HAZ: −1.33, $n = 238$). This analysis was restricted to cg18181703 as this is the only CpG covered by the Illumina 450 K array. We found a significant association with an increase of 0.14 cm per 1% increase in methylation ($P = 1 \times 10^{-3}$; see Table 3). Previous analyses of several of the cohorts studied here have reported associations between DNAm and maternal nutritional interventions[18] and season of conception (SoC)[18]. Sensitivity analyses confirmed that these cohort-specific exposures did not confound the observed *SOCS3m*−height associations (Supplementary Data 2 and 5).

To investigate whether the *SOCS3m*−height association observed in LMIC cohorts could be replicated in children from a HIC, we used data from the ALSPAC cohort ($n = 863$, UK). This analysis was restricted to cg18181703 as ALSPAC had methylation data available from the 450 K array only. A 1% increase in methylation was associated with a 0.11 cm increase in height, equivalent to 0.02 SD ($P = 2.2 \times 10^{-3}$; Table 3). The *SOCS3m*−height association was not influenced by cell composition effects (Supplementary Data 4).

**Table 1 | Cohort characteristics**

| | MMNP | MPC | PMMST | ENID (2 years) | ENID (6 years) | ALSPAC |
|---|---|---|---|---|---|---|
| **Child parameters** | | | | | | |
| Sample size (post QC) | 698 (685) | 557 (553) | 289 (284) | 239 (238) | 144 (142) | 925 (863) |
| Male (%) | 377 (55.0%) | 271 (49.0%) | 129 (44.6%) | 122 (51.2%) | 81 (57.0%) | 424 (49.2%) |
| Median age in years (IQR) | 5.80 (5.63–6.00) | 5.00 (4.98–5.01) | 9.00 (8.61–9.21) | 2.00 (2.00–2.01) | 6.20 (5.71–6.61) | 7.40 (7.23–7.57) |
| Mean height in cm (SD) | 109.60 (4.93) | 105.60 (4.31) | 130.00 (5.65) | 82.70 (3.03) | 113.00 (4.79) | 126.10 (5.20) |
| Mean height for age, HAZ (SD) | –1.01 (0.96) | –0.86 (0.91) | –0.33 (0.89) | –1.33 (0.96) | –0.66 (0.77) | 0.37 (0.93) |
| Stunted (%)[a] | 105 (15.3%) | 55 (9.9%) | 22 (7.7%) | 56 (23.5%) | 7 (4.9%) | 5 (0.6%) |
| Country (ethnicity) | India (SAS) | India (SAS) | Gambia (AFR) | Gambia (AFR) | Gambia (AFR) | United Kingdom (EUR) |
| **Maternal parameters** | | | | | | |
| Sample size (max N) | 683 | 553 | 276 | 238 | 142 | 863 |
| Mean BMI kg/m² (SD)[b] | 20.30 (3.73) | 23.60 (3.58) | 21.60 (3.62) | 21.16 (3.56) | 21.05 (3.05) | 22.90 (3.81) |
| Mean gestational age in weeks (SD) | 38.70 (2.18) | 39.10 (1.69) | 39.70 (2.88) | 39.70 (0.96) | 39.80 (0.91) | 39.60 (1.50) |
| Nutritional intervention (vs control) | Maternal MMN (Food-based) | None | Maternal MMN (Tablet-based) | Maternal /infant MMN/lipid/Fe-Fol supplementation | | None |
| DNA methylation platform[c] | EPIC | EPIC | EPIC | 450 K | EPIC | 450 K |
| Main cohort reference (PMID) | Potdar et al.[59] (25332324) | Krishnaveni et al.[20] (24609067) | Owens et al.[64] (26561613) | Moore et al.[21] (23057665) | Moore et al.[21] (23057665) | Fraser et al.[87] (22507742); Boyd et al.[22] (22507743) |

*MMNP* Mumbai Maternal Nutrition Project, *MPC* Mysore Parthenon Cohort, *PMMST* Periconceptional Multiple Micronutrients Supplementation Trial, *ENID* Early Nutrition and Immune Development, *ALSPAC* Avon Longitudinal Study of Parents and Children, *BMI* body mass index, *MMN* multiple micronutrients, *IQR* Inter quartile range, *QC* Quality Control based on meffil pipeline (see "Methods"), *SD* standard deviation, *HAZ* height for age, *PMID* PubMed ID, *SAS* South Asians, *AFR* Africans, *EUR* Europeans, *EPIC* Illumina Infinium methylation EPIC Beadchip, *450 K* Illumina Infinium methylation 450 K array.
[a]Stunting is defined as height <2 SD below WHO height-for-age reference mean.
[b]Maternal pre-pregnancy BMI except for MPC where it was measured at 28–32 weeks into pregnancy.
[c]All methylation measures are on genomic DNA isolated from peripheral blood samples collected at the same time as height measures.

## *SOCS3m* and stunting in LMIC cohorts

Stunting, defined as height <2 SD below the WHO height-for-age reference mean, is an important indicator of undernutrition and is observed predominantly in children from LMICs[13]. In the Indian cohorts, 10–15% of children were stunted, while in the Gambian cohorts stunting prevalence was higher at 2 years (24%) but much lower at 5–7 years (5–8%). To test whether *SOCS3m* is linked to stunting, we fitted logistic regression models with stunting as a binary outcome variable in all the LMIC cohorts. We identified an inverse association between stunting and *SOCS3m* at all three CpGs in the discovery cohort (MMNP) where a 1% increase in methylation was associated with an average 9.0% reduction in the relative risk of stunting (max $P = 1.4 \times 10^{-4}$, Table 4). The association was replicated in the MPC and ENID 2 years cohorts with similar effect sizes, but not in other Gambian (PMMST and ENID 6 years) cohorts which had lower rates of stunting and smaller sample sizes (Table 4).

## Assessing the influence of genetic variation on the *SOCS3m*–height association in LMIC

Methylation is influenced by genetic variation, notably in *cis*[23], and it is therefore possible that single nucleotide polymorphisms (SNPs) may confound methylation–height associations at *SOCS3*. Since we had limited power to detect genome-wide methylation quantitative trait loci (mQTL) in our datasets, we performed a *cis*-mQTL analysis in all the LMIC cohorts, restricted to SNPs within 1 Mb of the three *SOCS3* CpGs. No significant *cis*-mQTLs were identified for the three *SOCS3* CpGs (summarized in Supplementary Data 6). Next, to increase the power to detect *cis*-mQTLs, we performed meta-analysis for each CpG using the summary statistics from all four LMIC cohorts (Supplementary Data 7). Again, no significant mQTLs were identified, suggesting that *SOCS3m* is not influenced by *cis*-acting genetic variation. We additionally tested a cg18181703 trans-mQTL (rs4383852; chr6:109,594,475) previously identified in a large meta-mQTL analysis in Europeans[23]. This mQTL was

nominally associated with *SOCS3m* in PMMST only (*P* = 0.03) but it did not confound the main *SOCS3m*–height association in this cohort (95% CI: 0.00–0.07).

Since height is known to be strongly influenced by genome-wide genetic variation[6], we performed an additional sensitivity analysis using height polygenic risk scores (PRS) to test whether the *SOCS3m*–height association is independent of polygenic effects. For this analysis, individual height PRS were generated separately for each cohort using 12,111 SNPs identified in a large multi-ethnic study of adult height[6] (see Methods). *SOCS3* CpG effect sizes were unchanged after adjustment for PRS in all four LMIC cohorts, suggesting that *SOCS3m* predicts child height independent of genome-wide genetic influence (Supplementary Data 8).

## Assessing the influence of genetic variation on the *SOCS3m*–height association in HIC

Existing evidence supports the presence of ethnicity-specific genetic influences on both methylation[24] and height[25]. Since the high-income country cohort ALSPAC is of predominantly European origin, we investigated whether the *SOCS3m*–height association is confounded by genetic variation in this cohort and carried out further sensitivity analyses adjusting for the known cg18181703 trans-mQTL (rs4383852). Adjustment for rs4383852 genotype alone did not alter effect estimates. However, the association between cg18181703 and height was attenuated after adjustment for PRS (Supplementary Data 8).

## Proportion of height variance explained by *SOCS3m* and height PRS

We next investigated the proportion of height variance explained ($R^2$) by mean *SOCS3m* (across the three CpGs) and by height PRS in all cohorts using a single analysis of variance (ANOVA) model (see "Methods"). Similar amounts of height variance were explained by *SOCS3m* and height PRS in each LMIC cohort (Supplementary Data 9).

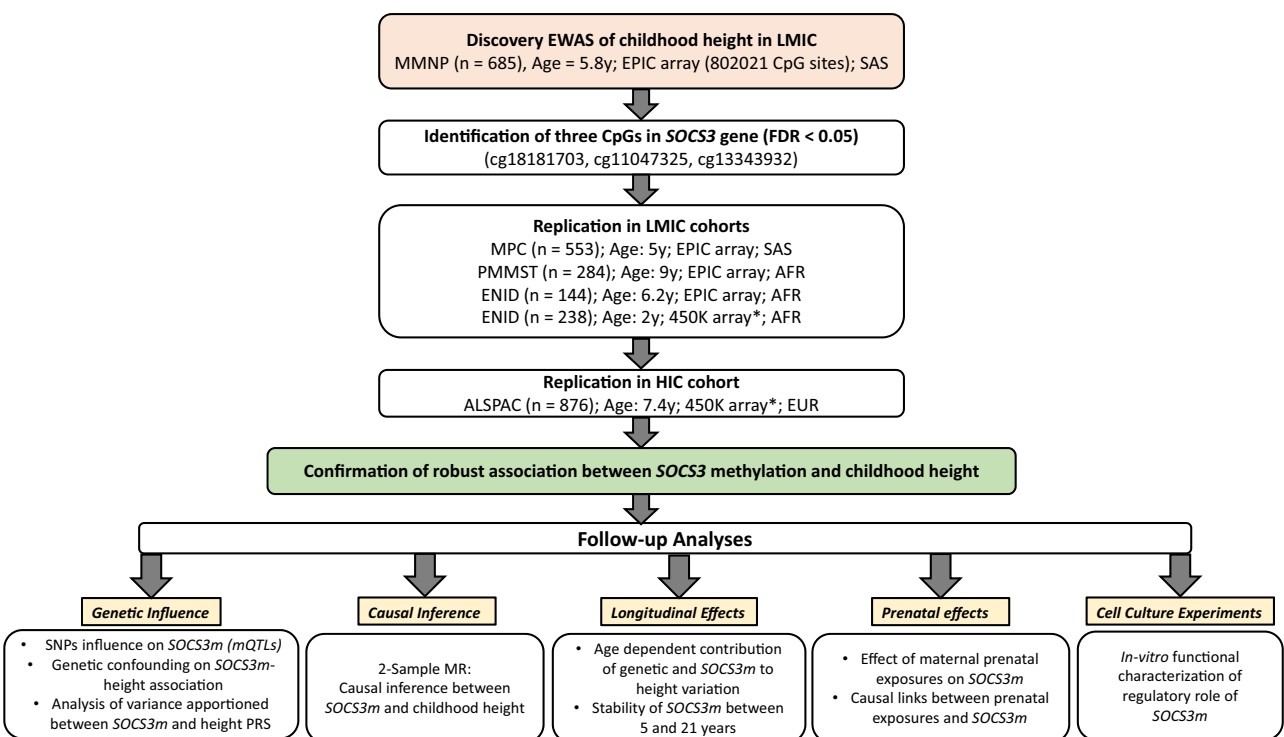

**Fig. 1 | Analysis workflow.** EWAS Epigenome-wide Association Study, LMIC low-middle-income countries, HIC high-income countries, MMNP Mumbai Maternal Nutrition Project, MPC Mysore Parthenon Cohort, PMMST periconceptional Multiple Micronutrients Supplementation Trial, ENID Early Nutrition and Immune Development, ALSPAC Avon Longitudinal Study of Parents and Children, SAS South Asians, AFR Africans, EUR Europeans, *SOCS3* Suppressor of Cytokine Signaling 3 gene, *SOCS3m* SOCS3 methylation, PRS Polygenic risk score, MR Mendelian randomization. *450 K array covers only one CpG (cg18181703). EPIC array: Illumina Infinium EPIC/850 K beadchip, 450 K: Illumina Infinium 450 K array. All analyses were cross-sectional unless indicated. All ages are medians.

**Table 2 | Discovery epigenome-wide association analysis of height in childhood in the MMNP cohort**

| CpG ID | *n* | Effect size[a] | 95% CI | *P* value | FDR | Chr | CpG position (hg19) | Gene name[b] |
|---|---|---|---|---|---|---|---|---|
| cg11047325 | 685 | 0.21 | 0.14, 0.28 | 3.0E−11 | 2.4E-05 | chr17 | 76,354,934 | *SOCS3* |
| cg13343932 | 685 | 0.25 | 0.18, 0.31 | 5.8E-11 | 4.6E-05 | chr17 | 76,355,061 | *SOCS3;LOC101928674* |
| cg18181703 | 685 | 0.30 | 0.24, 0.37 | 3.0E-10 | 2.4E-04 | chr17 | 76,354,621 | *SOCS3* |
| cg09383132 | 684 | 0.37 | 0.32, 0.42 | 2.3E-07 | 1.9E-01 | chr19 | 45,258,265 | *BCL3* |
| cg09050300 | 685 | −0.26 | −0.21, −.31 | 4.0E-07 | 3.2E-01 | chr5 | 95,511,892 | *LOC101929710* |
| cg19723657 | 685 | 0.15 | 0.10, 0.20 | 5.4E-07 | 4.4E-01 | chr13 | 92,002,951 | *MIR17HG;MIR20A;MIR19B1; MIR92A1;MIR18A;MIR19A* |
| cg24629020 | 685 | -0.14 | −0.09, −0.19 | 6.3E-07 | 5.1E-01 | chr12 | 15,752,531 | Intergenic |
| cg12170787 | 685 | 0.31 | 0.26, 0.36 | 1.5E-06 | 1.0E + 00 | chr19 | 1,130,965 | *SBNO2* |
| cg14972576 | 685 | 0.21 | 0.16, 0.25 | 2.8E-06 | 1.0E + 00 | chr5 | 97,645,526 | Intergenic |
| cg14472390 | 685 | −0.14 | −0.09, −0.18 | 2.9E-06 | 1.0E + 00 | chr5 | 140,740,387 | *PCDHGA4;PCDHGA2;PCDHGB2; PCDHGA1;PCDHGA3* |

*MMNP* Mumbai Maternal Nutrition Project, *CI* confidence interval, *FDR* false discovery rate, *Chr* chromosome, n sample size, *CpG ID* CpG probe on the Illumina Infinium methylation EPIC Beadchip array.
EWAS analysis was conducted using multiple linear regression models with a predefined FDR threshold of 0.05. The Benjamini–Hochberg method was used for multiple testing correction.
[a]Effect size indicates the change in height in centimetres associated with a 1% change in CpG methylation. The top three hits pass the FDR < 5% threshold for significance.
[b]UCSC reference gene name from EPIC manifest file. *P* value: statistical significance.
The top ten CpGs ranked by *P* value are shown.

In contrast, in the HIC ALSPAC cohort height variance explained by *SOCS3m* was much lower than that explained by the height PRS.

## Longitudinal height variance apportioned between *SOCS3m* and PRS

Existing evidence suggests that the relative genetic and environmental contributions to height variance are age-dependent, with genetic effects increasing with age and environmental exposures showing greater impact during early childhood[7]. To explore this, we performed a longitudinal analysis in the MPC cohort where height measures were available at different time points from birth to 21 years. We investigated the proportion of height variance from birth to adulthood, apportioned between *SOCS3m* measured at 5 years and the adult height PRS using combined ANOVA models (see "Methods"). *SOCS3m* and height PRS explained similar amounts of height variance throughout childhood and adolescence, but height PRS explained significantly more height variance than *SOCS3m* at 21 years (18% [11.6, 24.5] vs 1% [0.01, 3.02] respectively [95% CI]; Fig. 3 and Supplementary Data 10).

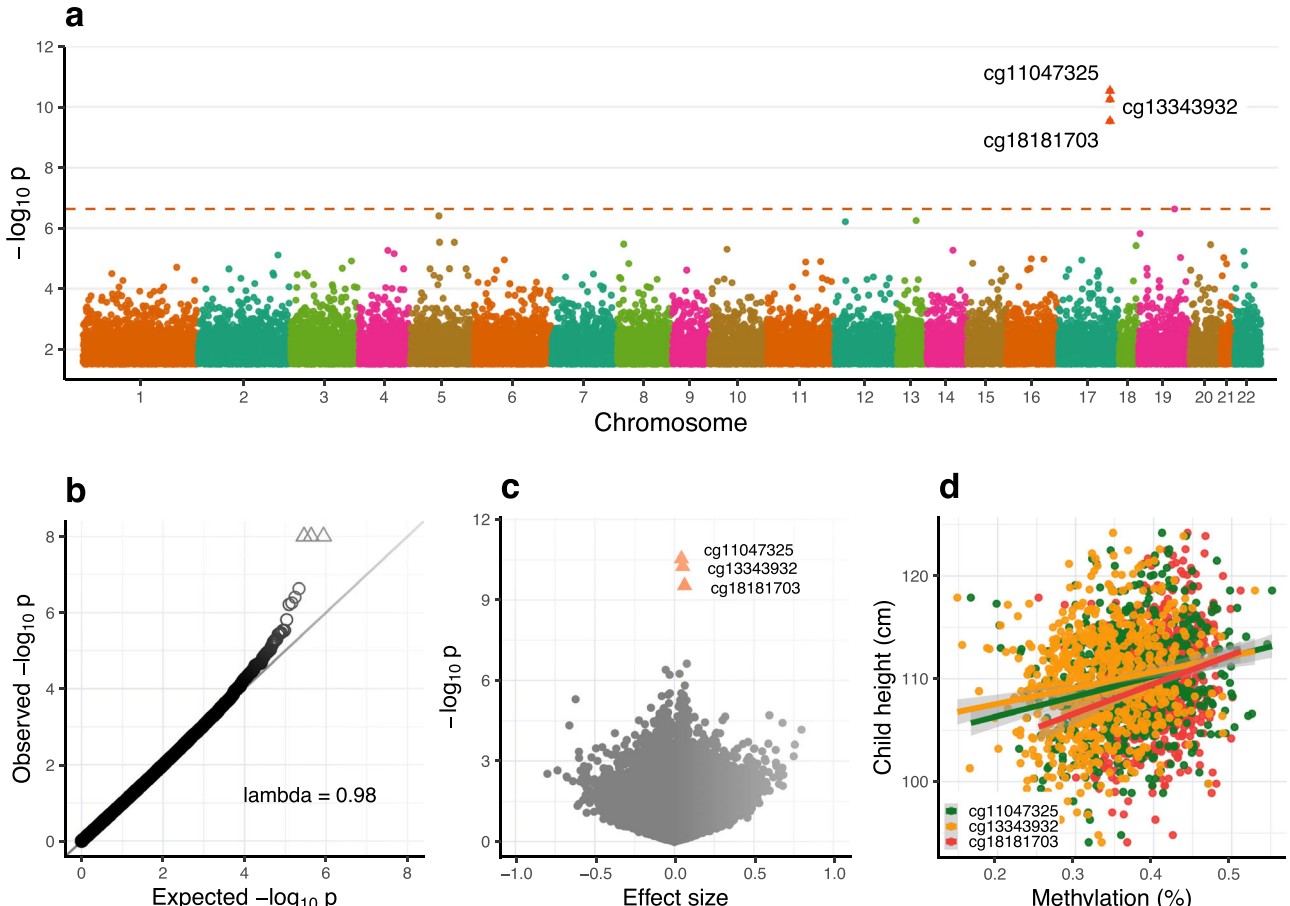

**Fig. 2 | Discovery epigenome-wide association analysis of height in children from the MMNP cohort.** **a** Manhattan plot showing epigenome-wide DNA methylation association results with respect to childhood height. Multiple linear regression models were used for the EWAS analysis. The dashed horizontal line represents Benjamini–Hochberg false discovery rate (FDR) = 0.05; $P = 2.4 \times 10^{-7}$). Arrowheads represent the three CpGs in *SOCS3* passing the FDR < 5% threshold. **b** Quantile–Quantile (Q–Q) plot for genomic inflation in *P* values. Lambda represents the genomic inflation factor. **c** Volcano plot showing effect sizes and *P* values. Arrowheads represent the FDR-significant CpGs in *SOCS3*. **d** Scatter plot showing effect sizes of significant CpGs. For each CpG, the gray-shaded area around the regression lines indicate 95% confidence interval for the estimated coefficient. EWAS Epigenome-wide association study.

**Table 3 | *SOCS3* methylation–height association in replication cohorts**

| Cohort | *n* | CpG ID | Effect size[a] | 95% CI | *P* value | Country (ethnicity) |
|---|---|---|---|---|---|---|
| MPC | 553 | cg11047325 | 0.20 | 0.15, 0.25 | 2.8E-14 | India (SAS) |
| | | cg13343932 | 0.21 | 0.16, 0.26 | 1.5E-13 | |
| | | cg18181703 | 0.29 | 0.21, 0.37 | 5.3E-12 | |
| PMMST | 284 | cg11047325 | 0.02 | 0.00, 0.04 | 4.7E-02 | Gambia (AFR) |
| | | cg13343932 | 0.02 | 0.00, 0.04 | 4.4E-02 | |
| | | cg18181703 | 0.04 | 0.00, 0.07 | 3.2E-02 | |
| ENID (6 years) | 142 | cg11047325 | 0.06 | 0.03, 0.10 | 1.4E-04 | Gambia (AFR) |
| | | cg13343932 | 0.08 | 0.04, 0.11 | 8.1E-05 | |
| | | cg18181703 | 0.11 | 0.06, 0.16 | 2.3E-05 | |
| ENID (2 years)[b] | 238 | cg18181703 | 0.14 | 0.06, 0.22 | 1.0E-03 | Gambia (AFR) |
| ALSPAC[b] | 863 | cg18181703 | 0.11 | 0.04, 0.18 | 2.2E-03 | United Kingdom (EUR) |

*MPC* Mysore Parthenon Cohort, *PMMST* Periconceptional Multiple Micronutrients Supplementation Trial, *ENID* Early Nutrition and Immune Development, *ALSPAC* Avon Longitudinal Study of Parents and Children, n sample size, *CI* confidence interval, *SAS* South Asians, *AFR* Africans, *EUR* Europeans, *CpG ID* CpG probe on the EPIC Infinium BeadChip Array.
Multiple linear regression models with predefined covariates were used for replication analyses (see "Methods").
[a]Effect size indicates change in height in centimetres associated with a 1% increase in CpG methylation.
[b]Methylation measures available from Illumina Infinium methylation 450 K array which covers only cg18181703. *P* value: statistical significance.

We also investigated associations between *SOCS3m* measured at 5 years and height across all available time points from 5 to 21 years in this cohort. The *SOCS3m*−height association was observed across all the time points, although there was some evidence that the strength of the effect decreased with age (Supplementary Data 11). Since height is strongly correlated across all time points, we performed an additional conditional height analysis, with height at each time point adjusted for all preceding height values, giving a measure of relative height gain at each time point.

**Table 4 | Association of *SOCS3* methylation with stunting in LMIC cohorts**

| Cohort | *n* | ᵃStunted *n* (%) | CpG ID | Effect size (OR)ᵇ | 95% CI | *P* value |
|---|---|---|---|---|---|---|
| MMNP | 685 | 105 (15.3) | cg11047325 | 0.92 | 0.89, 0.96 | 2.1E-05 |
| | | | cg13343932 | 0.93 | 0.89, 0.96 | 1.4E-04 |
| | | | cg18181703 | 0.89 | 0.84, 0.93 | 8.1E-06 |
| MPC | 553 | 55 (10.0) | cg11047325 | 0.93 | 0.89, 0.97 | 1.2E-03 |
| | | | cg13343932 | 0.93 | 0.89, 0.98 | 3.6E-03 |
| | | | cg18181703 | 0.91 | 0.85, 0.97 | 3.5E-03 |
| PMMST | 284 | 22 (7.7) | cg11047325 | 0.99 | 0.98, 1.00 | 1.2E-01 |
| | | | cg13343932 | 0.99 | 0.98, 1.01 | 2.2E-01 |
| | | | cg18181703 | 0.99 | 0.97, 1.01 | 2.8E-01 |
| ENID (6 years) | 142 | 7 (5.0) | cg11047325 | 0.89 | 0.79, 0.99 | 5.4E-02 |
| | | | cg13343932 | 0.89 | 0.78, 1.01 | 7.9E-02 |
| | | | cg18181703 | 0.86 | 0.71, 1.02 | 9.5E-02 |
| ENID (2 years) | 238 | 56 (23.5) | cg18181703 | 0.92 | 0.86, 0.99 | 2.8E-02 |

*LMIC* low- and middle-income countries, *MMNP* Mumbai Maternal Nutritional Project, *MPC* Mysore Parthenon Cohort, *PMMST* Periconceptional Multiple Micronutrients Supplementation Trial, *ENID* Early Nutrition and Immune Development, *ALSPAC* Avon Longitudinal Study of Parents and Children, *OR* odds ratio, *CI* confidence interval, n sample size.
ᵃStunting is defined as height <2 SD below WHO height-for-age reference mean.
ᵇLogistic regression estimates where 1% increase in methylation is related to [(1 − effect size in OR) * 100] percent of the reduction in the relative risk of stunting. Stunting was modeled by logistic regression with stunting as the binary outcome variable (see "Methods" for further details). *P* value: statistical significance.

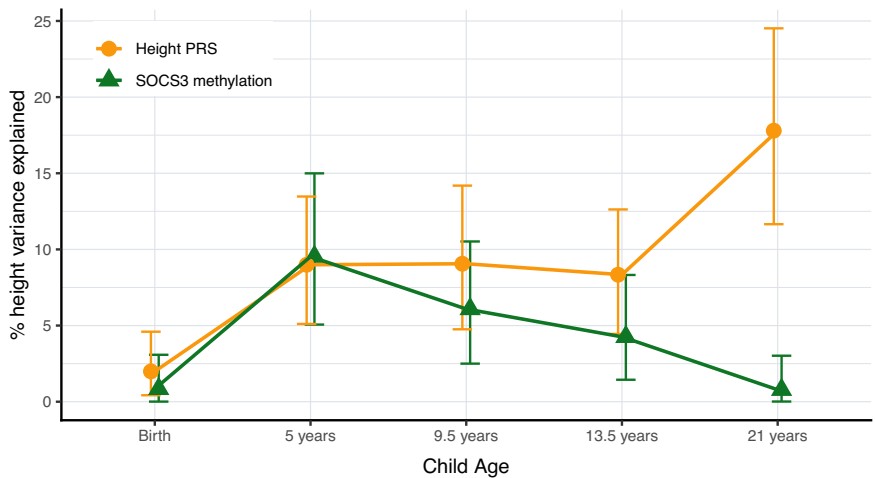

**Fig. 3 | Variance in height explained ($R^2$) by *SOCS3* methylation and PRS.** Longitudinal effects of height PRS and *SOCS3m* in the MPC cohort, showing percentage of variance apportioned between *SOCS3m* (mean methylation across all three *SOCS3* CpGs measured at 5 years) and height PRS across different time points from birth to early adulthood. *Y* axis shows the $R^2$ from ANOVA models including both mean *SOCS3m* and height PRS. Error bars indicate 95% confidence intervals for $R^2$ derived from *N* = 1000 bootstrap samples. *SOCS3m* *SOCS3* methylation, PRS polygenic risk score, ANOVA analysis of variance, LMIC low- and middle-income countries, HIC high-income countries, MPC Mysore Parthenon Cohort.

There was no significant association between *SOCS3m* at 5 years and height gained from 5 to 9.5 years or from 9.5 to 13.5 years. However, we found a significant inverse association between late childhood and adulthood, where increased *SOCS3* methylation at 5 years was associated with decreased height gain from 13.5 to 21 years (Supplementary Data 12).

### Stability of *SOCS3m*
Having identified the longitudinal effects of *SOCS3m* on height, we next assessed the stability of *SOCS3m* between childhood and adulthood by assessing correlation of DNAm values at 5 years (EPIC Array) and 21 years (Pyrosequencing) from the same 352 MPC participants. *SOCS3m* at all three CpGs was strongly correlated across these two time points (Supplementary Fig. 2), suggesting that *SOCS3m* is stable with age.

### Investigating the causal relationship between *SOCS3m* and height
To investigate the potential for a causal link between *SOCS3m* and height, we conducted a two-sample Mendelian randomization (MR) analysis using the previously identified European trans-mQTL (rs4383852; proxied by rs2884013) as an instrumental variable[26] and publicly available summary statistics from a large European height GWAS[27]. We found evidence of a causal effect of DNAm at cg18181703 on height with a 1% increase in methylation at cg18181703 associated with an 0.07 SD change in height (95% CI: 0.03–0.11; $P = 1.7 \times 10^{-3}$; Supplementary Data 13 and Supplementary Fig. 3).

### Effect of maternal pregnancy exposures on *SOCS3m*
Maternal factors during pregnancy can influence offspring DNA methylation[28]. We investigated the effect of various maternal exposures on *SOCS3m* in three LMIC cohorts where such data were available. Analyzed exposures included maternal body mass index (BMI), socioeconomic status (SES), pregnancy homocysteine, plasma/erythrocyte folate, and vitamin B12 concentrations (see "Methods" and Supplementary Data 14). Most maternal exposures did not show associations with *SOCS3m*, with the exception of maternal folate and

SES which were strongly replicated across Indian cohorts (Supplementary Data 15). In both the Indian cohorts, maternal folate and SES were significantly correlated (MMNP: Pearson's $r = 0.16$, $P = 8.9 \times 10^{-4}$; MPC: $r = 0.23$; $P = 2.2 \times 10^{-8}$). Since maternal folate status and SES could therefore be confounded, we further explored whether the folate and SES associations were independent in combined, multiple regression models. The results suggested that folate and SES are independently associated with *SOCS3m* (Supplementary Data 16).

Next, we carried out a MR analysis to investigate whether there was evidence of a causal relationship between maternal folate levels during pregnancy and *SOCS3m* in children at 5 years. We began by identifying SNPs within 1 Mb of the *MTHFR* gene which have previously been associated with folate levels to identify potential genetic instruments for maternal folate levels in Indian cohorts (see Methods). We identified four SNPs, and after LD filtering ($r^2 < 0.2$) we selected two coding variants in *MTHFR* (rs1801133 and rs2639453) as genetic instruments (Supplementary Data 17–19). MR analysis showed no evidence of a causal relationship between maternal folate exposure and *SOCS3m* in children at 5 years (Supplementary Data 20).

### Links between *SOCS3m* and BMI
*SOCS3m* has previously been linked to adult BMI[29]. We attempted to replicate this finding in one of our datasets (MPC) where cross-sectional data on both children and adults were available. In line with previous findings[30], we observed that *SOCS3m* was negatively correlated with BMI at 21 years. However, this effect was reversed in childhood (Supplementary Data 21). We performed sensitivity analyses and confirmed that the *SOCS3m*–BMI association at 21 years persisted after adjustment for height. However, no association was apparent at 5 years after adjustment for child height (Supplementary Data 21).

### In vitro functional characterization of height-associated *SOCS3m* region
The 441-bp region spanning the three identified *SOCS3* CpGs (chr17:76,354,621–76,355,061; hg19) is enriched for regulatory marks and overlaps a CpG Island (chr17:76,354,818–76,357,038; hg19) (Fig. 4a). Since methylation at CpG islands can influence gene expression[31], we investigated potential links between methylation and gene expression at the identified *SOCS3* region. We carried out in vitro luciferase assay experiments using a Lucia-based reporter vector (Supplementary Fig. 4) in human lung carcinoma (A549), human embryonic kidney (HEK293T), and human liver carcinoma (HepG2) cell lines. Compared to the basic vector, the *SOCS3* DMR constructs showed significant increase in the relative luciferase units (RLU) in A549 (RLU: 1.3; $P = 7.8 \times 10^{-8}$), HEK293 (RLU: 1.6; $P = 2.8 \times 10^{-6}$), and HepG2 (RLU: 5.8; $P = 1.3 \times 10^{-7}$) indicating enhancer activity at the cloned *SOCS3* region (Fig. 4b). We then investigated the role of methylation at this region by in vitro-methylation assays (see "Methods") in HepG2 cell line in both forward and reverse orientations. Methylation of the *SOCS3* insert attenuated the enhancer activity by 13% and 20% for forward and reverse orientations, respectively (Fig. 4c), indicating that DNA methylation at the cloned *SOCS3* region represses *SOCS3* expression.

## Discussion
Children in low and middle-income countries experience markedly different environmental conditions compared to their counterparts in high-income countries[32]. LMIC children are more frequently undernourished, experience higher rates of stunting and a greater burden of infection, all of which negatively affect their final height[12]. While several studies have investigated the genetic contribution to height variation, very few have probed the role of epigenetics, and to our knowledge none have analyzed the contribution of epigenetic factors to height differences in children from LMIC populations where the environmental contribution to height variance is greater than in HICs[33]. Here, we report results from an epigenome-wide DNA methylation analysis

of height in LMIC children. We identified a robust association between methylation at three CpGs within the second exon of the *SOCS3* gene and height in childhood. The observed epigenetic effect is independent of genetic factors, is consistent across four LMIC cohorts and is replicated in a UK HIC childhood cohort. Furthermore, we provide evidence of a causal effect of *SOCS3m* on height and of regulatory function associated with methylation of the *SOCS3* region.

Height is a classic example of a polygenic trait with thousands of genetic variants each contributing to height with small effect size[6] and the identification of specific genetic factors contributing to height heritability has been a long-standing topic of study in the field of human population genetics[6,27]. However, existing studies have focussed on adult height and it remains unclear whether identified SNPs also contribute to variance in child height. Our analysis suggests that *SOCS3m* independently accounts for a similar proportion of child height variance to cumulative genetic factors. In our discovery analysis, we found on an average 0.25 cm increase in height per 1% increase *SOCS3m*. This observed effect size at a single epigenetic locus is substantial when compared to the effect sizes of height-associated SNPs. The effect of *SOCS3m* in Indian cohorts (0.2–0.3 cm per 1% increase in methylation) was comparatively larger than that observed in the Gambian cohorts which might reflect differences in environmental exposures.

Existing evidence suggests that the relative genetic contribution to stature increases with age, while environmental factors have their greatest effect during early childhood[7]. In the MPC, which has height data from birth to young adulthood, the association between *SOCS3m* and height was positive throughout (Supplementary Data 11) but the effect size diminished with age. Our analysis using conditional height variables as indices of height gain in successive discreet age intervals (Supplementary Data 12), showed that *SOCS3m* was only strongly associated with height gain between birth and 5 years. These findings suggest that *SOCS3m* could be a mediator of environmental effects on height during early childhood. A negative association of *SOCS3m* with height gain between 13 and 21 years (Supplementary Data 12) opens the possibility that it advances skeletal maturation, shifting more height gain to earlier ages and reducing pubertal height gain. This requires exploration in other cohorts.

We observed that the height variance explained by PRS and *SOCSm* were similar in LMIC cohorts and that methylation effects were independent of cumulative genetic factors. In contrast, in the HIC cohort analysis the PRS explained much greater height variance compared to *SOCS3m*. Given evidence of a causal relationship between *SOCS3m* and child height, these results suggest that epigenetic modifiers may have a greater effect on child growth in LMIC, while genetic factors are major determinants of height in HIC children. In our analysis we used a PRS derived from a large multi-ethnic GWAS on adult height in 5.4 million individuals[6]. In their analysis, Yengo et al. speculate that the reduced phenotypic variance explained by the PRS in non-Europeans could be due to unidentified genetic variants specific to non-European ancestries, and/or to a greater effect of non-genetic factors. We are unable to determine the relative contributions of either of these in our study, although our analysis of the effect of prenatal factors linked to *SOCS3m* suggests that the early environment may play a role.

Prenatal environmental exposures can influence the offspring epigenome[16], and studies in The Gambia have highlighted associations between DNA methylation, season of conception and maternal nutrition[18,34]. Maternal folate deficiency during pregnancy has been previously linked to stunting, and antenatal folate supplementation in mothers has been shown in some studies to improve linear growth in the children[35]. We found that both maternal pregnancy folate and socioeconomic status were independently associated with *SOCS3m*, although we were unable to demonstrate a causal link between maternal folate and *SOCS3m*. Further studies with larger sample sizes could establish whether such a link exists.

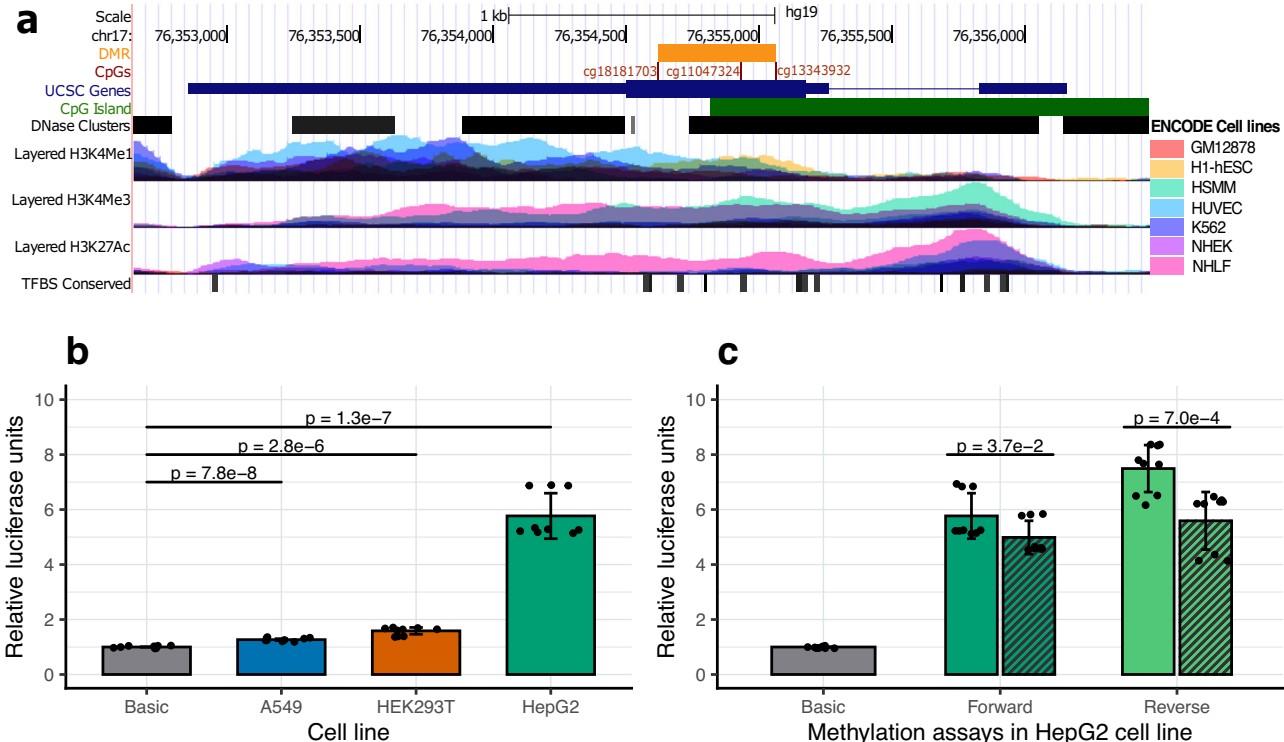

**Fig. 4 | In vitro functional characterization of *SOCS3* methylation. a** UCSC genome browser plot showing the *SOCS3* gene (blue bar), CpGs (red lines), and DMR (orange bar) identified in the EWAS. The tracks from top to bottom indicate a CpG island (chr17: 76,354,818–76,357,038; hg19); DNase hypersensitivity clusters showing open chromatin (125 cell types from ENCODE version 3); histone tail marks H3K4Me1, H3K4Me3 and H3K27Ac indicating active regions of transcription, and transcription factor binding sites (ENCODEv3). Overlayed colored tracks for the histone tail marks represent seven cell lines from ENCODE, including GM12878: Lymphoblastoid cell line, H1-hESC human embryonic stem cell line, HSMM skeletal muscle myoblast cell line, HUVEC human umbilical vein endothelial cell line, K562 chronic myelogenous leukemia cell line, NHEK normal human epidermal kerati-nocyte cell line, NHLF normal human lung fibroblast cell line. **b** Luciferase assay experiments using CpG free Lucia vector showing enhancer activity of the *SOCS3* DMR constructs (forward orientation) in human lung carcinoma (A549), human embryonic kidney (HEK293T) and human liver carcinoma (HepG2) cell lines. **c** Luciferase assay experiments in HepG2 cell lines showing a regulatory role of methylation at the *SOCS3* DMR. Methylated (striped bars) versus unmethylated *SOCS3* constructs are compared in both forward (dark green) and reverse (light green) orientations. **b, c** Normalized mean relative luciferase units (RLU) from three experiments are shown on the *y* axis and the error bars indicate standard deviation of RLU. Two-sided *t* tests were used for all comparisons. UCSC University of California Santa Cruz, DMR Differentially Methylated Region, EWAS Epigenome-Wide Association Study.

Evidence from animal models suggests that exposures in early gestation can influence postnatal phenotypes through epigenetic mechanisms and there is speculation that similar effects exist in humans[36]. We found that *SOC3m* is stable between the ages of 5 and 21 years but we were unable to determine whether *SOC3m* states are established in prenatal or early life. However, if *SOCS3m* were to be established during early embryonic or fetal development, we can speculate that *SOCS3m* might influence postnatal growth in response to early nutritional and/or other environmental factors giving rise to an inter-generational effect.

Short stature in childhood has been linked to multiple adverse outcomes in later life including increased risk of type 2 diabetes[37], cardiovascular disease[38], and obesity[39], and *SOCS3* expression has also been associated with similar disease phenotypes in adults[40–42]. Hypo-methylation at cg18181703 is associated with a 1.38-fold increased risk of obesity[42] and promoter methylation of *SOCS3* is associated with metabolic syndrome[43]. cg18181703 methylation is also linked to inflammation through its association with peripheral blood CRP levels[44] and fluctuating *SOCS3* levels have been linked to the inflam-matory disease rheumatoid arthritis[45]. Further studies are required to explore the links between childhood *SOCS3m*, stunting and inflammation-related outcomes in later life.

Studies in adults have found associations between *SOCSm* methylation and BMI[46]. A meta-EWAS of BMI by Wahl et al.[30] showed that *SOCS3*m at cg18181703 is negatively associated with BMI in adults,

with evidence that DNAm changes are driven by changes in adiposity. We replicated this association in MPC at 21 years, but not in early childhood, and confirmed that the association in adults is not con-founded by height. This suggests that *SOCS3m* may be linked to adult BMI and height through different biological pathways.

The ubiquitously expressed *SOCS3* gene belongs to the family of suppressor of cytokine signaling (SOCS) genes and is a predominant feedback inhibitor of the JAK/STAT pathway[47]. Evidence suggests that *SOCS3*-mediated inhibition of the JAK/STAT pathway is critical for bone formation[48] and normal skeletal development in humans[49]. Dys-regulation of the JAK/STAT/SOCS axis has been found to inhibit bone formation and lead to skeletal abnormalities including shortened limbs at birth[50], abnormal bone formation and reabsorption[51], and stunting in childhood[52]. As linear growth during childhood is crucial for deter-mining final adult height, our finding of a link between *SOCS3m*, child height and child stunting in populations with a high prevalence of stunting is notable. *SOCS3* is also a major regulator of immune responses involved in anti/pro-inflammation and plays a role in mod-ulating the outcome of infections[53]. Abnormal expression of *SOCS3* has been implicated in chronic inflammatory disease[54]. Chronic inflam-mation is associated with stunting in LMIC[55] and it negatively impacts linear bone growth[56].

Histone modification marks provide further insights into the *SOCS3* region's regulatory role. We found evidence of enhancer activity and active transcription in several cell lines using public data on

histone marks from ENCODE. Furthermore, our in vitro experiments demonstrated that this region can act as a gene enhancer with methylation attenuating enhancer activity. This is in line with observations that methylation at cg18181703 is negatively correlated with *SOCS3* expression in hepatocellular carcinoma, arthritis, schizophrenia, and chronic hepatitis-B[43]. Gene body methylation is generally linked to differential expression of gene isoforms[57]. *SOCS3* has two cytokine-inducible isoforms (long and short) with varying expression levels (Supplementary Fig. 5b), half-lives and stability[58]. As enhancers are known to regulate isoform expression and DNAm can influence enhancer activity, variable levels of methylation at this region might influence preferential expression of *SOCS3* isoforms. Further studies are needed to investigate whether *SOCS3m* leads to isoform-specific *SOCS3* expression implicated in differential inflammatory responses.

We note that this study was limited to loci on the Illumina EPIC and 450 K arrays and that a genome-wide analysis would improve the ability to detect epigenetic factors associated with child height. Other limitations include a lack of available methylation data at birth so that we were unable to investigate cross-sectional associations with birth length or perinatal exposures. As noted above, the height PRS used in this study may have lower prediction accuracy in non-European populations compared to Europeans, although we note that the PRS is derived from a sample that includes a substantial number of individuals of non-European ancestry. While our results are in line with previous observations[7] that the effect of genetic variation increases from childhood to adulthood, using PRS generated from large non-European GWAS studies may give better estimates of the relative contribution of PRS to height.

In summary, the genetic architecture of height is well-characterized in adults but the relative contributions of genetic and epigenetic factors on child height are under-explored, particularly in low and middle-income countries where epigenetically mediated environmental effects may play a greater role. We have identified a novel, robust association between methylation in the second exon of the *SOCS3* gene and height in children from LMIC and HIC cohorts, with evidence of a causal link. *SOCS3* is implicated in diverse physiological processes relating to bone development, metabolism and inflammation. Further work is required to establish the molecular pathways linking the epigenetic regulation of *SOCS3* expression to child height.

## Methods

### Study cohorts

The participants in this study were children from five cohorts from India, The Gambia, and the United Kingdom. The Indian cohorts (South Asians) comprised the Mumbai Maternal Nutritional Project (MMNP) and the Mysore Parthenon Cohort (MPC). The Gambian cohorts included children from the Preconceptional Multiple Micronutrients Supplementation Trial (PMMST) and Early Nutrition and Immune Development (ENID) trial with African ancestry. The Avon Longitudinal Study of Parents and Children (ALSPAC) is a UK (European) cohort recruited in the United Kingdom and represents the only HIC cohort in the study. In all cohorts', informed consent was obtained from the parents of the participating children. Further details on individual cohorts are given below. A summary of cohort characteristics is provided in Table 1.

**Mumbai Maternal Nutrition Project.** The MMNP (ISRCTN62811278, also known as Project SARAS [meaning "excellent"]) was a randomized controlled trial (RCT) of a food-based supplementation containing multiple micronutrients conducted among women living in slum communities in Mumbai, India[59]. The supplementation was a daily snack made from local micronutrient-rich foods, which started preconceptionally and continued until delivery. The aim of this study was to investigate whether improving the mothers' diet quality pre-

conceptionally and throughout pregnancy by increasing intake of micronutrient-rich foods improved offspring birth outcomes. The children were followed up at the age of 5–7 years for various health outcomes, including height anthropometry as part of the SARAS KIDS study[60]. In the current study, child height was measured to the nearest millimeter at the age of 5–7 years, using a wall-mounted stadiometer (Microtoise, CMS Instruments Ltd., UK) and the average of three such measures was considered. Peripheral blood samples were collected at the same age and used for profiling DNA methylation. Pre-pregnancy maternal BMI was calculated following standard methods at the time of recruitment. Maternal plasma folate was measured by microbiological assay using samples collected at 7–16 weeks gestation and stored at −80 °C using a chloramphenicol-resistant strain of *L. casei*[61] and plasma cobalamin (B12) was measured by microbiological assay using a colistin sulfate-resistant strain of *L. leichmanii*[62]. Socioeconomic status in this cohort was derived using Standard of Living Index (SLI) questionnaire, developed for India's National Family Health Survey. The SLI questionnaire creates a score based on the size and quality of housing and amenities and ownership of land and household assets, with a higher score reflecting a higher SES[63].

**Mysore Parthenon Cohort.** The MPC was a prospective study set up in 1997–1998 in Mysore, South India, to assess the prevalence of gestational diabetes and long-term effects of maternal nutritional status during pregnancy on cardiovascular and cognition outcomes in their children[20]. Detailed anthropometry of children born to these women were collected at the time of follow-up at 5 years of age (*n* = 585), 9.5 years (*n* = 539), 13.5 years (*n* = 545) and 21 years (*n* = 352). Maternal folate, homocysteine and vitamin B12 levels were analyzed in the third trimester (28–32 weeks of gestation) using plasma samples stored at -80ºC at the Diabetes Unit, KEM hospital, India. Folate and vitamin B12 were measured following microbial assays and homocysteine was measured using fluorescence polarization immunoassay (Abbott Laboratories, Abbott Park, IL, USA). The current study involved measurement of blood DNA methylation by Illumina Methylation EPIC Bead Chips (EPIC array) at 5 years and pyrosequencing at 21 years.

**Periconceptional multiple micronutrients supplementation trial.** PMMST (ISRCTN13687662) was a periconceptional micronutrient supplementation trial conducted between 2006 and 2008 in The Gambia in Sub-Saharan West Africa. The purpose of the trial was to investigate the effect of preconceptional micronutrient supplementation on placental function[64]. Children born to PMMST mothers were followed up at 7–9 years as a part of the EMPHASIS study which included measurement of child height and blood DNAm measured on the Illumina EPIC array[60].

**Early nutrition and immune development.** ENID (ISRCTN49285450)[21] was a partially blinded randomized controlled trial conducted in rural Gambia to investigate the effect of prenatal and infant nutritional supplementation on infant immune development. Women were recruited into the study from 2010 to 2014 during early pregnancy (10–20 weeks into gestation) and received either (i) iron-folate (standard care); (ii) multiple micronutrients (MMN); (iii) energy, protein, and lipid with iron-folate; or (iv) energy, protein, and lipid with MMN supplements until delivery. Infants were further randomized to receive lipid-based nutritional supplements, with or without additional micronutrient supplementation from 6 to 18 months of age. Maternal BMI and plasma nutritional biomarkers, including homocysteine, folate and B12 were measured on samples collected at the time of recruitment, before supplementation (median 14 weeks gestation)[21]. Blood samples and anthropometric measures including child height were obtained from a subset of ENID children aged 2 years as part of a study identifying biomarkers and understanding mechanisms for the

relationship between aflatoxin exposure and child stunting[65]. DNA was extracted from white blood cells and genome-scale methylation profiles were obtained using the Illumina Human Methylation 450 K Bead Chips (450 K array) as described in a previous publication[66]. A follow-up study measured child heights and generated DNA methylation data using the Illumina EPIC array in a subset of ENID children at 5–7 years of age[67,68]. In the current study, we have used child height at 2 years and 6 years for association studies.

**Avon longitudinal study of parents and children.** Pregnant women resident in Avon, UK with expected dates of delivery between April 1, 1991 and December 31, 1992 were invited to take part in the ALSPAC study[22]. The initial number of pregnancies enrolled was 14,541. Of these initial pregnancies, there were a total of 14,676 foetuses, resulting in 14,062 live births and 13,988 children who were alive at 1 year of age. Methylation data was generated in a subset of mother-child pairs using 450 K Illumina arrays[69]. Methylation data used in this study was generated using blood samples collected when children were ~7 years old.

Child height was measured to the last complete mm using the Harpenden Stadiometer at clinic visits attended when children were ~7 years old. Maternal and partners ethnic group were obtained by questionnaires administered to mothers during pregnancy. Non-white individuals were removed from the dataset ($n < 5$). A total of 925 children had methylation data and were included in the dataset used in this study. Children with missing height ($n = 46$) or genetic data ($n = 16$) were removed from analyses. The study website (http://www.bristol.ac.uk/alspac/researchers/our-data/) contains details of all the data that is available through a fully searchable data dictionary and variable search tool.

### Ethics and consent

The EMPHASIS study was registered as ISRCTN14266771. The Ethics approval for the EMPHASIS study in India was obtained from the Intersystem Biomedical Ethics Committee, Mumbai in 2013 (serial no. ISBEC/NR-54/KM/JVJ/2013) and in The Gambia from the Joint Gambia Government/MRC Unit The Gambia's Ethics Committee in 2015 (Serial no. SCC 1441). EMPHASIS was a follow-up study of the MMNP and PMMST trials. MMNP was approved by the BYL Nair and TN Medical College, Sir JJ Group of Hospitals, and Grant Medical College, Mumbai, India. PMMST was approved by the Joint Gambia Government/Medical Research Council (MRC) Unit The Gambia's Ethics Committee (L2005.111v2 SCC 1000). MPC was approved by the Holdsworth Memorial Hospital (HMH) research ethics committee, Mysore, India. The ENID trial was approved by the joint Gambia Government/MRC The Gambia Ethics Committee (SCC1126v2). Ethical approval for the ALSPAC study was obtained from the ALSPAC Ethics and Law Committee and the Local Research Ethics Committees. Ethical clearance for processing biological samples and generation of molecular biology data was obtained from the Institutional Ethics Committee of CSIR-CCMB, Hyderabad (No. IEC-41/2015). Consent for biological samples in ALSPAC has been collected in accordance with the Human Tissue Act (2004). The study was conducted in accordance with criteria set out in the Declaration of Helsinki.

### Pre-processing of child height measures

Height measures for each cohort were first assessed for normality, and outliers were removed before generating residuals. As the height of an individual varies with age and sex, we first generated height residuals adjusted within each cohort for the child's age and sex by linear regression. The residuals were centered to have mean 0 and variance 1. Child stunting, defined as standing height-for-age Z-score (HAZ) more than two standard deviations below the WHO growth reference mean was computed using the *"addWGSR"* function from the R package *"zscorer"* (version 0.3.1).

### DNA methylation profiling and data processing

Detailed information related to sample processing, methylation data generation and quality control for MMNP[18], PMMST[18], ENID[66], and ALSPAC[69–71] have been previously published. We provide some further brief details below.

Peripheral blood samples from individuals were collected at the time of follow-up, plasma was separated and packed cells were stored in EDTA vacutainers at −80 °C until DNA isolation. DNA isolations in each cohort were carried out separately using QIAamp blood DNA isolation kit from Qiagen following manufacturers' protocols. DNA quality and quantity were estimated by Invitrogen Quant-iT PicoGreen (Invitrogen) and/or Nanodrop 1000 Spectrophotometer (Thermo Fisher Scientific, USA). Samples with good DNA quality were then bisulfite converted using Zymo EZ DNA methylation kits, (Zymo Research, Irvine, CA). The converted DNA were then processed either on the EPIC array (Illumina Inc., San Diego, USA−MMNP, PMMST and ENID 6-year cohorts), or on the 450 K array (ENID 2 year and ALSPAC). Arrays were scanned on an Illumina iScan and the first pass quality review was carried out using GenomeStudio. Raw *.idat* files were imported into *R* and processed using the *R* Bioconductor package *"meffil"* with default parameters[72]. CpG probes identified as cross-reactive[73] or mapping to X/Y chromosomes were removed. Finally, the data from all cohorts were independently normalized using Functional Normalization as a part of the *meffil* pipeline[74].

For MPC, genomic DNA from 5-year-old children ($n = 561$) were randomized on the EPIC chip using the Optimal Sample Assignment Tool (*OSAT*) *R* package and DNAm data was generated on the EPIC array using the above-mentioned protocol. On QC analysis, no sex mismatches were identified. One sex detection outlier (>5 SDs from the mean), and six outliers with predicted median methylation of >3 SDs from the regression line were excluded. Finally, probes with detection *P* values > 0.01 ($n = 2985$) and/or bead numbers <3 in more than 10% of the samples were excluded. Following *meffil* default parameters, functional normalization was carried out using the first 20 PCs derived from the control probes and the final dataset comprised 862,933 probes and 554 samples.

### Generation of high-throughput genotype data and imputation

Genome-wide SNP data from the Illumina Infinium Global Screening Array-24 v1.0 Beadchip (GSA array; Illumina, California, USA) was available for MMNP, PMMST and MPC. Detailed information on data processing has been reported previously[18]. Briefly, quality control involved the removal of samples with genotyping call rate <95% and SNPs with Hardy-Weinberg equilibrium (HWE) *P* value > $5 \times 10^{-6}$. For imputation, array-derived genotypes were pre-phased using SHAPEITv2[75] and imputation was carried out using IMPUTE2 software (version 2.3.2)[76] using the 1000 Genomes phase 3 reference panel. SNPs with a minor allele frequency (MAF) < 0.10 and an IMPUTE2 "info" metric <0.9 were excluded, the latter to ensure maximum confidence in imputation quality.

For ENID, genotypes were obtained using the Illumina H3Africa array which has a high representation of African genomic variation. Genotype data was filtered using PLINKv1.90b6.24 with the options −geno 0.05 −hwe 1e-6 −maf 0.05. Haplotype reconstruction was then performed using SHAPEIT v4.2.2 with the option −window 5 and using the b37 genetic maps provided in the SHAPEIT repository. IMPUTE5 v1.1.5 was then used to chunk and impute the phased genotypes, with 1000 Genomes Phase 3 (1000G) genotypes serving as the reference. The 1000G hap and legend files were converted to VCF format using PLINK2 v2.00a3.3LM with the ref-first option set. The chunking was done using the "imp5Chunker" command with options −window-size 5,000,000 and −buffer-size 500,000. Then the imputing was performed on each chunk using the "impute5" command with the −out-gp-field flag set and the options −r and −buffer-region set according to the output of the chunking command. The imputed genotypes were

then filtered using the bcftools 1.9 command 'view' with option –include "INFO/INFO > = 0.9" for each chunk, and the chunks for each chromosome were then combined into a single BCF file using bcftools concat with options -n -f as recommended by the IMPUTE5 user manual. The BCF files were converted to bed/bim/fam format using PLINK, and PLINK then used to merge the chromosome-level files into a single dataset with the options –keep-allele-order –make-bed –merge-list.

ALSPAC children were genotyped using the Illumina HumanHap550-quad chip. SNPs with MAF < 0.01, missingness >0.05, and HWE $P$ value $< 1 \times 10^{-6}$ were excluded. Individuals were further excluded on the basis of sex mismatches, minimal or excessive heterozygosity, disproportionate levels of individual missingness (>3%) or insufficient sample replication (identity by descent; IBD < 0.8). Population stratification was assessed by multidimensional scaling analysis and individuals with non-European ancestry were removed. Phasing was conducted using SHAPEITv2 and imputation was carried out using IMPUTE2 (version 2.2.2) using the 1000 genomes phase 1 version 3 reference panel. SNPs with less than 0.8 information matrix score were excluded.

### Prenatal exposures

Data on multiple maternal exposures measured pre-pregnancy or during pregnancy were available for all LMIC cohorts except for PMMST. Analyzed exposures comprised maternal BMI, socioeconomic status, pregnancy folate and vitamin B12 (all cohorts); gestational diabetes (MMNP and MPC only); homocysteine (MPC and ENID); and vitamin D (MPC only). Further details on maternal exposures are summarized in Supplementary Data 14. Analysis of associations with offspring *SOCS3m* was carried out using linear regression models using *glm()* in *R* with methylation residuals preadjusted for child age, sex and batch effects as the outcome and maternal exposure as the independent variable. Associations were considered significant at $P < 0.05$.

### Pyrosequencing

Genomic DNA from blood samples at 21 years from the MPC cohort were pyro-sequenced to assess the stability of *SOCS3m* between 5 and 21 years of age. Only samples that overlapped the 5-year samples with methylation measured on the EPIC array were selected. The assays were designed using the PyroMark Assay Design Software (ver. 2.0.1.15), and sequencing was performed using the PyroMark Q96 MD pyrosequencer (both Qiagen, Hilden, Germany). 400 ng of genomic DNA from 406 participants was subjected to bisulfite conversion using the EZ-96 DNA Methylation-Goldkit (Zymo Research). Two PCR products were amplified, a 150 bp amplicon for cg18181703 and a 246 bp amplicon for cg11047325 and cg13343932 and pyrosequencing was caried out using three sequencing primers in 96-well plates (Supplementary Data 22). Non-methylated (0%) and fully methylated (100%) standards (Qiagen, Hilden, Germany) were used as controls to examine the consistency of methylation levels across all batches. The data were then assessed for call rates and quality using Pyromark q96 SW2.0 software.

### In vitro functional characterization of the height-associated region in *SOCS3*

The genomic region encompassing the three *SOCS3* CpGs (cg18181703, cg11047325, and cg13343932) was PCR amplified (495 bp) and cloned into the Lucia vector (Supplementary Fig. 4) using *ApaI* and *BamHI* restriction enzymes (NEB, Catalog no. R0114S and R0136S, respectively). The vector harbors no CpG dinucleotides making it suitable for measuring the effect of methylation exclusively from the cloned *SOCS3* fragment. Details on the oligo primers used for cloning and a vector map are provided in Supplementary Data 23. In vitro methylation of the plasmid with and without the *SOCS3* insert was carried out using M.SssI methylase enzyme (NEB, cat. no: M0226S) following standard protocols and successful methylation of the insert was confirmed by

digestion with methylation-sensitive restriction enzymes *AfeI*, *BsaHI*, *HgaI*, and *BstU1* (New England Biolabs, USA) and Sanger sequencing.

The cell culture experiments were carried out in three cell lines A549 (source: Lung), HEK293 (Kidney) and HepG2 (Liver) to identify a robust regulatory signal associated with *SOCS3* region. These cell lines were chosen based on gene expression levels in the GTEx database (https://gtexportal.org/home/) covering varying equivalent tissue expression levels (Supplementary Fig. 5a) and on relevant literature[77,78]. The Cell lines were grown in Dulbecco's Modified Eagle Medium (DMEM) supplemented with 10% fetal bovine serum and 1% (50 U/mL) of antibiotics (penicillin and streptomycin, Thermo fisher scientific) under standard cell culture conditions. Cells were seeded at a density of 150 K cells per well in 24-well culture plates (Thermo Fisher Scientific) and grown until 70–80% confluency prior to transfections. Recombinant plasmids were transfected into cell lines using Lipofectamine 2000 (Invitrogen; cat.no: 11668019). Vector backbone and unmethylated constructs were used as internal controls for comparison. Luciferase readings were recorded 36 h post-transfection using the Dual luciferase reporter assay kit (Promega; cat. no. E1910) on Perkin Elmer's Multimode plate reader (Perkin Elmer, Catalog no. 6055400). The gene expression measures were obtained as relative luciferase units and fold changes were calculated by comparing luciferase readings between methylated and unmethylated constructs. Student *t* tests were performed to estimate the statistical significance of differential expression levels.

### Statistical analyses

**Discovery epigenome-wide association study.** The discovery EWAS was carried out in the MMNP cohort which comprised 685 individuals with methylation measured at 803,210 CpG loci on the EPIC array that passed QC. The EWAS was conducted using the *R* package "ewaff" (version: 0.0.2; https://github.com/perishky/ewaff). To account for batch effects, surrogate variables (SVs) were derived from the 200 K most variable CpG probes. A generalized linear regression model was used with height residuals as the outcome (dependent variable) and methylation beta values (methylation values from 0 to 1) as exposure (independent variable) while adjusting for child age, child sex, and ten methylation-derived SVs. Significant loci were defined as those with a Benjamini–Hochberg false discovery rate (FDR) of <0.05 to account for multiple testing. Inflation was assessed by computing the genomic inflation factor (*lambda*) and inspecting quantile–quantile (Q–Q) plots. The standardized estimates from the regression output were transformed to height in centimetres for ease of interpretation. A separate sensitivity EWAS analysis with direct blood cell counts (neutrophils, lymphocytes, eosinophils, basophils and monocytes) as additional covariates was conducted to detect any potential influence of variability in blood cell proportions.

An additional genome-wide analysis of differentially methylated regions (DMR) associated with height was carried out using the "*DMRcate*" package with the default parameters of window size: $\lambda = 1000$, scaling factor for bandwidth: C = 2. Significant DMRs were defined as those passing Stouffer (adjusted $P$ value < 0.05).

**Replication analysis.** We used three independent child cohorts from LMICs and one from an HIC to test for replication of the CpG loci associated with mid-childhood height in the discovery analysis. LMIC cohorts comprised an additional Indian cohort (MPC, $n = 553$), two Gambian cohorts (PMMST, $n = 284$; ENID 6 years: $n = 144$), and the HIC cohort was from the UK (ALSPAC, $n = 863$).

CpG–height associations were estimated using regression models similar to the ones used in the discovery analysis. Briefly, generalized linear regression models were fitted for each cohort, with height residuals as the outcome variable and methylation as the exposure variable. Child age and sex plus batch effects including slide, sentrix row and/or processing batch were included as covariates in the model.

Comparable effect sizes were obtained by converting standardized estimates to height in centimetres and a nominal significance level of $P < 0.05$ were considered evidence for replication. In each cohort additional sensitivity analyses to assess the potential influence of variation in blood cell proportions were conducted using cell count estimates generated using the Houseman method[79]. Previous studies in Gambian cohorts[18,34] showed an effect of season of conception (SoC) on offspring DNAm. We therefore conducted sensitivity analyses for SoC effect on *SOCS3m*–height association in Gambian cohorts using Fourier (cosinor) regression models as previously described[34]. All analyses were carried out in *R* (version 4.1.2).

**Association of *SOCS3m* with stunting.** Logistic regression models were fitted with stunting as a binary outcome variable and *SOCS3m* as the exposure, adjusted for child age, sex, methylation batch, slide and sentrix row. For ease of interpretation regression coefficients were converted to odds ratios which are interpreted as odds of having the outcome per unit (%) change in methylation.

***SOCS3m* association with BMI**. Potential links between *SOCS3m* and BMI were assessed using linear regression models with BMI residuals adjusted for sex and age as the outcome and *SOCS3m* as the exposure variable. Methylation-specific technical variables including batch, slide and sentrix row were included as covariates.

**Assessment of genetic influence on *SOCS3m*–height association.**

 i. *Methylation quantitative trait locus (mQTL) analysis*

 To test for the potential influence of genetic variation on methylation, we conducted an mQTL analysis using the "G-model" in the "*GEM*" package (v1.10.0)[80] separately for each cohort. *Cis*-SNPs within 1 Mb of the CpGs of interest were considered in this analysis. An additive (allelic dose) model was used with CpG methylation residuals (preadjusted for child age and sex) as the dependent variable and SNP genotypes encoded as 0, 1, 2 as the independent variables. There were ~2000 SNPs within the specified window in all the cohorts. The exact number of SNPs tested in each cohort are summarized in Supplementary Data 6. Significant *cis*-mQTLs were those with an association *P* value passing a Bonferroni corrected significance threshold of $P = 0.05$/number of SNPs and CpGs tested. To increase the power to detect SNP-methylation associations, we conducted an additional meta-analysis on the mQTL summary statistics using all four LMIC cohort following the inverse-variance method in METAL software (version release 2011-03-25)[81].

 ii. *Polygenic Risk Score (PRS)*

 A height PRS provides a cumulative measure of genome-wide genetic variation that predicts an individual's height. The height PRS was generated using 12,111 autosomal height-associated SNPs identified by the GIANT consortium using a conditional and joint multiple-SNP (COJO) analysis[6]. We used the effect sizes of the height-increasing allele for each SNP identified in the COJO analysis as weights and the individual PRS was calculated as follows:

$$PRS_j = \sum W_i \times G_{ij}$$

Where, $PRS_j$ is polygenic score of individual j, "$W_i$" is the weight (effect size) of $SNP_i$ taken from the COJO analysis and $G_{ij}$ is the genotype at $SNP_i$ in individual j, coded as 0, 1 or 2 based on the height increasing allele. The function --score in PLINKv1.9 was used to generate a PRS score for each individual. The final PRS was calculated by multiplying the score and the number of non-missing alleles from the output of the --score function.

**Investigating the causal relationship between *SOCS3m* and child height by Mendelian randomization (MR) analysis.** We used publicly available GWAS summary statistics on height[27] ($n = 253,288$) and mQTL associated with DNA methylation at cg18181703 identified in a recent large mQTL meta-analysis ($n = 27,746$; http://mqtldb.godmc.org.uk/search?query=cg18181703)[23] in a two-sample MR. This was conducted using the "*TwoSampleMR*" R package where height was the outcome, and methylation was the exposure (see Supplementary Fig. 3). Under these assumptions, MR provides estimates of the causal effect of the exposure on the outcome[82].

Genetic instruments for DNA methylation were obtained from the GoDMC database by extracting all *cis* variants with a *P* value $< 1 \times 10^{-8}$ and *trans* variants ($>1$ Mb from CpG site) with a *P* value $< 1 \times 10^{-14}$ (see ref. 23). After linkage disequilibrium (LD) clumping using parameters of LD $r^2 = 0.001$ and 10 Mb LD windows, one mQTL (rs4383852) for cg18181703 was identified. This SNP (rs4383852) was missing in the outcome GWAS dataset and was therefore proxied by an alternative SNP (rs2884013) in complete LD with the first ($r^2 = 1.0$)[83]. SNP alleles were harmonized to represent a single effect allele so that both exposure and outcome variants were expressed per effect allele increase for the exposure. Since a single mQTL was identified as a proxy for the exposure of interest, the Wald ratio estimator was used to quantify the causal effect. Using the Wald ratio, we calculated the change in height per unit increase in methylation, where the numerator is the SD change in height and the denominator is the increase in methylation % per copy of the effect allele, respectively.

**Analysis of height variance explained by *SOCS3m* and PRS**. ANOVA was used to estimate the height variance independently explained by *SOCS3m* and height PRS. The height variance explained by methylation and PRS was assessed in a single regression model, including mean methylation of the three *SOCS3* CpGs, height PRS, child age, sex, and methylation batch variables. The variance explained ($R^2$) by individual variables in the final regression models was obtained using the "*Anova()*" function from the "*car*" package (version 3.1-0). For each variable in the model, the $R^2$ can be interpreted as the percentage of the variance in height explained by that variable. For instance, the $R^2$ for a predictor variable in the model is the sum of squared differences in height attributed to that predictor variable divided by the sum of squares for all the variables in the model. Confidence intervals for $R^2$ were calculated using "boot" package with 1000 bootstrap replications.

**Selection of SNPs and causal inference analysis between prenatal exposures and *SOCS3m*.** Genetic variants in and around the *MTHFR* gene are known to be associated with folate levels. We therefore restricted our search for folate-associated genetic instruments to within 1 Mb of *MTHFR* in the GWAS catalog (https://www.ebi.ac.uk/gwas/). Genetic association analysis was conducted in Indian cohorts where both serum folate levels and genetic data were available: specifically, MPC mothers and children, MMNP mothers, and Pune Maternal Nutrition Study (PMNS)[84] mother, father and child, and individuals in Mysore Birth Record Cohort (MBRC)[85]. Association analysis was performed by linear regression using an additive model with the minor allele count as an independent variable and the residual standardized $\log_{10}$ transformed folate levels adjusted for age, BMI, and sex as the dependent variable using PLINKv1.9 (Supplementary Data 17). Fixed-effect inverse-variance weighted meta-analysis using the summary statistics from all cohorts identified four SNPs, rs375679568, rs3737967, rs2274976, and rs1801133 in the *MTHFR* gene to be associated with folate levels with a GWAS significant threshold of $P = <5 \times 10^{-8}$ (Supplementary Data 17). The first three SNPs were in strong LD ($r^2 > 0.9$) in South Asians (1000 Genome Phase 3 data, Supplementary Data 18). We therefore selected rs2274976 (Arg594Gln) and the fourth variant rs1801133 (Ala222Val) as genetic instruments for folate exposure in MR analysis. We next performed

independent association analysis in MMNP and MPC cohorts between maternal rs2274976 and rs1801133 variants, and *SOCS3* methylation levels in the children, and meta-analyzed the results as above (Supplementary Data 19). The resulting effect estimates and standard errors were used in a two-sample MR with an inverse-variance weighted formula to understand the effect of maternal folate exposure on *SOCS3m* in offspring[86].

## Reporting summary

Further information on research design is available in the Nature Portfolio Reporting Summary linked to this article.

## Data availability

Requests to access the MMNP data should be submitted to K. Kumaran, Giriraj R. Chandak, and for the MPC data to G.V. Krishnaveni. Requests to access the Gambian data should be submitted to the corresponding authors in the first instance. An application would then need to be made to MRC Unit The Gambia's Scientific Coordinating Committee and the Joint MRC/Gambia Government Ethics Committee. EMPHASIS study data is available on request and will be made publicly available once results from the main EMPHASIS study have been published. ALSPAC data used for this submission is available on request by application to the ALSPAC executive committee (ALSPAC-exec@bristol.ac.uk). The ALSPAC data management plan (available here: www.bristol.ac.uk/alspac/researchers/access/) describes in detail the policy regarding data sharing, which is through a system of managed open access. All relevant data supporting the key findings of this study are available within this article and its Supplementary Data files. Publicly available data used in this study include GoDMC (http://mqtldb.godmc.org.uk/search?query=cg18181703), GWAS catalog (https://www.ebi.ac.uk/gwas/), ENCODE (https://www.encodeproject.org/) and GTex (https://gtexportal.org/home/) databases. Source data are provided with this paper.

## Code availability

All analyses used open-source software including R (Version 4.1.2), PLINK (version 1.90b6.24; v2.00a3.3LM), IMPUTE2 (version 2.3.2), SHAPEIT (version 4.2.2), METAL (version release 2011-03-25), zscorer (version 0.3.1), meffil (version 1.3.6), DMRcate (version 2.14.0), GEM (version 1.10.0), and TwoSampleMR (version 0.5.6).

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

## Acknowledgements

MMNP: We are grateful to the families who took part, and the team of fieldworkers, nurses, research assistants, and data managers who carried out the study. The original trial was supported by the Wellcome Trust, United Kingdom; the Medical Research Council, United Kingdom; the Department for International Development, United Kingdom; USAID; the Parthenon Trust, Switzerland; and the Industrial Credit and Investment Corporation of India Bank Ltd Social Initiatives Group, Mumbai, India. The SARAS KIDS children's follow-up study was funded by the Medical Research Council, UK research grant no: MR/M005186/1. PMMST and ENID: We thank all the study participants in West Kiang, The Gambia for their time and commitment. We also thank members of the laboratory and field teams. We acknowledge the work of Z. Herceg, M.N. Routledge, Y.Y. Gong, and H. Hernandez-Vargas in acquiring the ENID 2-year 450k array data and Toby Candler in acquiring samples and data in the follow-up study in 5–7-year-old children. PMMST was supported by MRC (U1232661351, U105960371, and MC-A760-5QX00) and DFID under the MRC/DFID Concordat, and other members of the Gambian team were supported by MRC grants U105960371, U123261351 and MR/M01424X/1. The ENID trial was jointly funded by the UK Medical Research Council (MRC) and the Department for International Development (DFID) under the MRC/DFID Concordat agreement (MRC Program MC-A760-5QX00). Methylation analysis of ENID early childhood samples was supported by the Bill & Melinda Gates Foundation (grant no. OPP1 066947). The generation of methylation data from the same cohort in late childhood was funded by the UK MRC (grant no. MC_PC_MR/RO20183/1). The EMPHASIS study was jointly funded by MRC, DFID, and the Department of Biotechnology (DBT), Ministry of Science and Technology, India, under the Newton Fund initiative (MRC Grant No.: MR/N006208/1 and DBT Grant No.: BT/IN/DBT-MRC/DFID/24/GRC/2015–16). MPC: We thank all the participating families, CSI Holdsworth Memorial Hospital staff, the research team, and the staff of the MRC Lifecourse Epidemiology Unit for their support. MPC was supported by the Parthenon Trust, Switzerland; the Wellcome Trust, UK; the MRC, UK and the Department for International Development (UK). The follow-up at 21 years was supported by the DBT/Wellcome Trust India Alliance Fellowship [Grant Number: IA/CPHS/16/1/502655]. Thanks to the Council of Scientific and Industrial Research (CSIR), Ministry of Science and Technology, Government of India, India for funds for generating high-throughput genomic and methylation data. ALSPAC: We are extremely grateful to all the families who took part in this study, the midwives for their help in recruiting them, and the whole ALSPAC team, which includes interviewers, computer and laboratory technicians, clerical workers, research scientists, volunteers, managers, receptionists, and nurses. Hannah R. Elliott, Eleanor C.M. Sanderson, and Caroline L. Relton are members of the Medical Research Council Integrative Epidemiology Unit at the University of Bristol, which is supported by the Medical Research Council and the University of Bristol (MC_UU_00011/5 and MC_UU_00011/1). The UK Medical Research Council and Wellcome (Grant ref: 217065/Z/19/Z) and the University of Bristol provide core support for ALSPAC. A comprehensive list of grants funding is available on the ALSPAC website (http://www.bristol.ac.uk/alspac/external/documents/grant-acknowledgements.pdf). GWAS data was generated by Sample Logistics and Genotyping Facilities at Wellcome Sanger Institute and LabCorp (Laboratory Corporation of America) using support from 23andMe.

## Author contributions

C.H.D.F., G.R.C., M.J.S., and K.A.L. conceived and designed the study. M.J.S., P.I., H.R.E., S.S.N., K.A.L., G.R.C., and C.H.D.F. devised the analysis strategy; P.I., H.R.E., S.S.N., A.S., N.J.K., and M.D. conducted the statistical and bioinformatics analysis. H.R.E., E.S., and K.A.W. devised and conducted the ALSPAC cohort MR analysis; K.K., S.A.S., G.V.K., S.E.M., and A.M.P. provided phenotype data for the Indian and Gambian cohorts, and C.D.G. processed the EMPHASIS study phenotype data; C.H.D.F., S.A.S., R.D.P., H.C., H.S., and M.G. set up the original trials in India and A.M.P., M.J.S., L.J., A.P., and S.H.K. set up the original trials in The Gambia; Sa.S., S.R.M., P.I., S.B., Sm.S., and M.B. performed the methylation assays; M.A., P.I., and G.R.C. designed cell culture assays and M.A. conducted the experiments; P.I., M.J.S, H.R.E., S.S.N, M.A., G.R.C., and C.H.D.F. drafted the manuscript. K.A.L., E.S., S.E.M., A.M.P., S.O., and C.L.R. have critically revised the first draft, and all authors reviewed and approved the final manuscript.

## Competing interests

The authors declare no competing interests.

## Additional information

## the EMPHASIS study group

Prachand Issarapu [1,2], Manisha Arumalla[1], Smeeta Shrestha [1], Sara Sajjadi[1,5], Chiara Di Gravio[9], Sirazul A. Sahariah[10], Ramesh D. Potdar[10], Harsha Chopra[10], Harshad Sane[10], Meera Gandhi[10], Landing Jarjou[2], Ann Prentice[2], Sarah H. Kehoe[8], Stephen Owens[14], Caroline L. Relton [3,4], Andrew M. Prentice [2], Karen A. Lillycrop[12,13], Matt J. Silver[2] ✉, Giriraj R. Chandak [1,5] ✉ & Caroline H. D. Fall[8]

[14]Institute of Health and Society, Newcastle University, Newcastle, UK.

