## [Peer Review File · Nature Communications]

DNA methylation at the suppressor of cytokine signaling 3 (SOCS3) gene influences height in childhoodREVIEWER COMMENTS

Reviewer #1 (Remarks to the Author):

This is a very nicely written, clear and scientifically interesting study that I found very enjoyable to read. The analysis is comprehensive and continues a series of nice studies from these authors. Results are very compelling and will be of strong interest to the field internationally.

There are several strengths of this study that support biological relevance of the findings;

- Comprehensive and appropriate analytical approach
- Strong discovery evidence
- Independent replication across multiple datasets (robust finding)
- Additional link to stunting
- Tested for evidence of mQTL effect, including with height PRS (only minor evidence of a role for genetics in some cohorts)
- Causal inference work (causal role)
- Additional molecular evidence (pyrosequencing)
- In vitro functional work (functional relevance at a molecular level) using a CpG free reporter system

Queries:

L114: Fig4 shows CpG sites within exon 1 of SOCS3, whereas this states 'These three CpGs mapped to intron 2 of Suppressor of cytokine signaling 3 gene (SOCS3) on chromosome 17'

L361: evidence of enhancer extends to the profile of chromatin modifications in the region. This could be further explained and the potential for tissue specific enhancer activity ascertained from the public encode data

Reporter work: rationale for choosing lung (A549), blood (HEK293T), liver (HEPG2) derived cell lines should be included

L680: the authors downplay the role of mQTL (genetic) effects in the observed relationship between changes in SOCS3 methylation and height, yet use an mQTL linked to methylation at cg18181703 (their main interest) in MR analysis. This could be explained a little more clearly. Additionally, there are genetic data available for some of these studies and a direct examination of the relationship between height and rs2884013 genotype in

Reviewer #2 (Remarks to the Author):

My comments are:

1. Can the authors comment on the effect of DNA methylation of these CpGs on the relative expression of the isoforms of SOCS3?
2. On page 5, the authors state "These three CpGs mapped to intron 2 of Suppressor of cytokine signaling 3 gene (SOCS3) on chromosome 17" while in all other places it is mentioned as exon 2. Which one is correct?

Issarapu et al response to reviewers

We thank the reviewers for taking the time to evaluate our work and for their insightful comments.

Reviewer #1 (Remarks to the Author):

This is a very nicely written, clear and scientifically interesting study that I found very enjoyable to read. The analysis is comprehensive and continues a series of nice studies from these authors. Results are very compelling and will be of strong interest to the field internationally.

There are several strengths of this study that support biological relevance of the findings;

- Comprehensive and appropriate analytical approach
- Strong discovery evidence
- Independent replication across multiple datasets (robust finding)
- Additional link to stunting
- Tested for evidence of mQTL effect, including with height PRS (only minor evidence of a role for genetics in some cohorts)
- Causal inference work (causal role)
- Additional molecular evidence (pyrosequencing)
- In vitro functional work (functional relevance at a molecular level) using a CpG free reporter system

We are pleased to note the reviewer's positive comments.

Queries:

Comment1 (C1): L114: Fig4 shows CpG sites within exon 1 of SOCS3, whereas this states 'These three CpGs mapped to intron 2 of Suppressor of cytokine signaling 3 gene (SOCS3) on chromosome 17'

Response 1 (R1): We thank the reviewer for drawing our attention to this error. As shown in Figure 4, the identified *SOCS3* CpGs are within exon 2, as we reported throughout the manuscript except at the line highlighted by the review. The error has been corrected in the revised manuscript (L114).

C2: L361: evidence of enhancer extends to the profile of chromatin modifications in the region. This could be further explained and the potential for tissue specific enhancer activity ascertained from the public encode data.

R2: We assume the reviewer is referring to the *SOCS3* region histone modifications illustrated in Figure 4a which are indicative of enhancer activity (H3K27ac, H3K4me1) and active transcription (H3K4me3). We agree with the reviewer and have further explained the origins and significance of these marks in the Figure 4a caption and in the Discussion (L362-364).

C3: Reporter work: rationale for choosing lung (A549), blood (HEK293T), liver (HEPG2) derived cell lines should be included

R3: The cell lines were chosen based on evidence of gene expression from published literature (Ruscica M et al 2016, Duan W et al 2017) with further insights from the tissue expression data base GTEx (<https://gtexportal.org/home/>). The rationale was to choose cell lines from different tissues that express *SOCS3* and also to cover cell types that range in levels of gene expression to robustly replicate the findings. We have further explained our rationale for cell line selection in the revised manuscript (L595-599). We have also provided the *SOCS3* multi-tissue expression profile from GTEx as a supplementary figure (Sup. Figure 5a).

C4: L680: the authors downplay the role of mQTL (genetic) effects in the observed relationship between changes in *SOCS3* methylation and height, yet use an mQTL linked to methylation at cg18181703 (their main interest) in MR analysis. This could be explained a little more clearly.

R4: These were two different analyses with different aims.

In our first set of analyses, we investigated whether genetic effects could be confounding the observed *SOCS3m*-height association. Our mQTL analysis (L161) sought to identify SNP-*SOCS3* CpG associations through separate and combined (meta-analysis) of all the LMIC cohorts and found no evidence of genetic effects. Since power was limited with this analysis, we performed an additional analysis (L174-180) to test whether the *SOCS3m*-height association was independent of polygenic effects using the height PRS from Yengo et al. Again, we found no evidence of confounding suggesting that *SOCS3m* predicts child height independent of genome-wide genetic influence. Finally, we conducted a further test in the ALSPAC cohort which is of predominantly European ancestry using a known European trans-mQTL. Again, we found no evidence of confounding (L186).

In a second MR analysis, we used the same European trans-mQTL (rs2884013) that was identified in a large European study (GoDMC; n~200K) as a genetic instrument to investigate the causal relationship between height and *SOCS3m* using statistics from a large height GWAS in Europeans (L222; L686).

We have further clarified that the MR analysis was conducted using European data only in the manuscript (L224-226).

C5: Additionally, there are genetic data available for some of these studies and a direct examination of the relationship between height and rs2884013 genotype in

R5: The reviewer's comment ends in mid-sentence. We assume that the reviewer is suggesting that we test the relationship between height and the European trans-mQTL (rs2884013) using genotype data available in the study cohorts.

We can think of two reasons for doing this: i) to test the hypothesis that our main findings are confounded by this trans-mQTL observed in Europeans only; and/or ii) to test the hypothesis that *SOCS3m* mediates an effect of the trans-mQTL on height.

We have investigated the first hypothesis in Europeans (ALSPAC) in the current manuscript (L183-187). We have now additionally tested for confounding in the Gambian PMMST cohort where this mQTL also shows a significant association with *SOCS3m* (L172-173).

We have investigated the second causal hypothesis in Europeans (L226) since we only have European GWAS data.

Reviewer #2 (Remarks to the Author):

Comment 1: Can the authors comment on the effect of DNA methylation of these CpGs on the relative expression of the isoforms of SOCS3?

Response 1: The *SOCS3* gene has two isoforms [ENST00000330871.3, ENST00000587578.1; RefSeq genes from NCBI, Annotation Release NCBI Homo sapiens 105.20220307 (2022-03-12)]. We were unable to identify any studies integrating isoform expression and DNA methylation in the *SOCS3m* region. We have now elaborated on the plausible role of *SOCS3* methylation on isoforms expression in the Discussion (L370-374) and we have included a figure (Supplementary Figure 5b) showing *SOCS3* isoform expression profiles across different tissues for more clarity.

C2: On page 5, the authors state “These three CpGs mapped to intron 2 of Suppressor of cytokine signaling 3 gene (SOCS3) on chromosome 17” while in all other places it is mentioned as exon 2. Which one is correct?

R2: We apologize for this typo. This is now corrected in the revised manuscript.

REVIEWERS' COMMENTS

Reviewer #1 (Remarks to the Author):

the authors have addressed my comments satisfactorily

Reviewer #2 (Remarks to the Author):

The revised manuscript has addressed all the reviewer comments.

REVIEWERS' COMMENTS

Reviewer #1 (Remarks to the Author):

the authors have addressed my comments satisfactorily.

Reviewer #2 (Remarks to the Author):

The revised manuscript has addressed all the reviewer comments.

We are very much pleased to note the reviewer's positive comments and we thank both the reviewers for taking time to review our manuscript.